# Enhancing Integrated Gradients Using Emphasis Factors and Attention for Effective Explainability of Large Language Models

## Abstract

Understanding the decision-making processes of large language models (LLMs) is critical for ensuring transparency and trustworthiness. While Integrated Gradients (IG) is a popular method for model explainability, it faces limitations when applied to autoregressive models due to issues like exploding gradients and the neglect of the attention mechanisms. In this paper, we propose an enhanced explainability framework that augments IG with emphasis factors and attention mechanisms. By incorporating attention, we capture contextual dependencies between words, and the introduction of emphasis factors mitigates gradient issues encountered during attribution calculations. Our method provides more precise and interpretable explanations for autoregressive LLMs, effectively highlighting word-level contributions in text generation tasks. Experimental results demonstrate that our approach outperforms standard IG and baseline models in explaining word-level attributions, advancing the interpretability of LLMs.

## 1 Introduction

As large language models (LLMs) become increasingly prominent in natural language processing tasks (Kenton & Toutanova (2019); Jha et al. (2020)), understanding their decision-making processes is critical for ensuring transparency and trustworthiness Lipton (2018). Autoregressive models, in particular, generate text by predicting one word at a time based on the preceding context, making it essential to interpret how individual words influence subsequent predictions. Traditional model explainability techniques, such as Integrated Gradients (IG), have been widely used to quantify the contribution of input features to model outputsShrikumar et al. (2017); Lundberg (2017); Murdoch et al. (2018). However, when applied to autoregressive models, IG faces inherent challenges due to their sequential nature, often leading to inaccurate or incomplete explanations Enguehard (2023). Further related works has been discussed in Appendix A.1. In autoregressive text generation, capturing the contextual dependencies between words is crucial for reliable interpretability Vaswani (2017). Moreover, common challenges, such as exploding gradients during the gradient calculation for long texts using the IG method, further complicate the task of identifying meaningful token-level contributions. To address these challenges, we propose an enhanced explainability framework integrating attention mechanisms and emphasis factors with IG. Attention allows us to account for the relationships between words in the context window while scaling factors mitigate the gradient-related issues that can obscure proper explanations. We make the following key contributions in this paper:

1. We identify the limitations of the exploding gradient problem when applying the Integrated Gradients (IG) method for attribution analysis for long texts using generative LLMs.

2. We propose a novel solution to address the exploding gradient problem encountered during attribution calculations in the Integrated Gradients method.

3. We integrate the Attention mechanism into the attribution calculation, as it plays a critical role in predicting the next token in large language models (LLMs).

4. We conduct a comprehensive comparative study, evaluating our proposed method against several baseline models across multiple datasets and architectures.

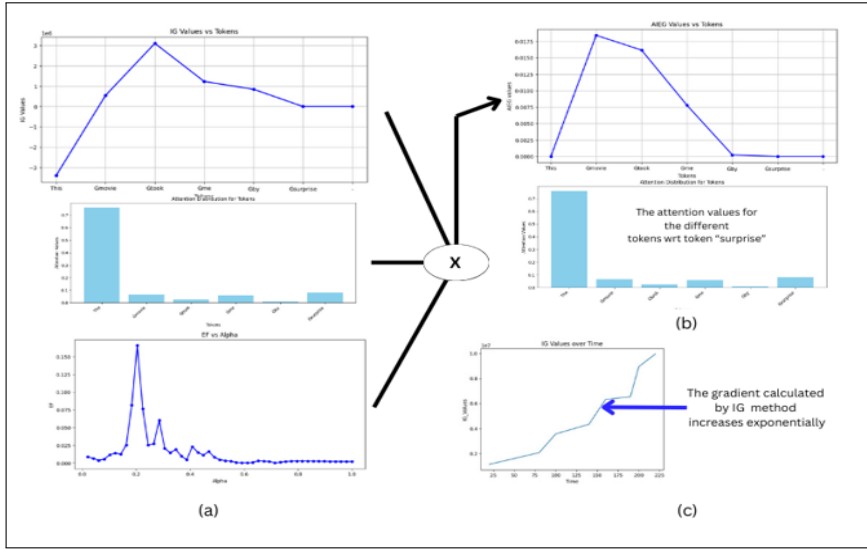

Figure 1: (a) shows the IG values, the masked self-attention values and the EF values of the tokens with respect to the "surprise" token for the text "The movie took me by surprise." produced by the GPT2-small model. Our method combines these to create the values of AIEG method. (b) shows the self-attention values with respect to the token "surprise".(c) shows the accumulation of gradient of the output word with respect to a particular input word from the beginning of the text, over time in the Integrated Gradient method as the length of the generated text gets long.

## 2 LIMITATIONS OF GRADIENTS AS ATTRIBUTIONS FOR GENERATIVE MODELS

**Axiom: Sensitivity:** The gradient-based method does not satisfy the sensitivity axiom. Let's demonstrate this with a straightforward example using a simple RNN. These are the general equations for the hidden state and output for an RNN, as given below.

$$\text{Hidden State Update:} \quad h_t = \sigma(\mathbf{W}_{hx} \cdot x_t + \mathbf{W}_{hh} \cdot h_{t-1} + \mathbf{b}_h) \tag{1}$$

$$\text{Output Calculation:} \quad y_t = \phi(\mathbf{W}_{hy} \cdot h_t + \mathbf{b}_y) \tag{2}$$

The hidden state at time step $t$ is denoted by $h_t$. The activation function, denoted by $\sigma$ and $\phi$, is typically a non-linear function such as $\tanh$ or ReLU. which introduces non-linearity and affects gradient flow during backpropagation. The trainable weight matrix for the input $x_t$ is represented by $\mathbf{W}_{hx}$. Here, $x_t$ denotes the input at time step $t$. The trainable weight matrix for the previous hidden state $h_{t-1}$ is given by $\mathbf{W}_{hh}$, and $h_{t-1}$ is the hidden state from the previous time step. $\mathbf{b}_h$ and $\mathbf{b}_y$ represent the bias terms. Now we will try to create a simple RNN from equations (1) and (2). We take $\mathbf{W}_{hx} = -1, \mathbf{W}_{hh} = 0, \mathbf{W}_{hy} = 1, \mathbf{b}_h = 1, \mathbf{b}_y = 1, h_{t-1} = 0$, and $\phi = \sigma = \text{ReLU}$. So the equation becomes: $h_t = \text{ReLU}(1 - x_t)$ and $y_t = \text{ReLU}(1 + h_t)$. The value of $y_t$ is 2 and 1 for values of $x_t = 0$ and 2 respectively. The value of $x \cdot \frac{\partial y_t}{\partial x}$ is 0 for both the values of $x_t$, indicating that sensitivity is not preserved.

## 3 APPROACH USING INTEGRATED GRADIENTS

In the paper Sundararajan et al. (2017), the authors demonstrate the application of Integrated Gradients to neural machine translation models utilizing LSTM architectures. They compute the contribution of each input token to the probability of every output token, which is represented in the form of wordpieces. This process effectively aligns the output sentence with the input sentence. For the baseline, the authors set the embeddings of all tokens to zero, except for the start and end markers. Suppose for a neural network $F$, the goal is to compute attributions IG($x$) that quantify the contribution of each input word to the network's output. Consider an input $x \in \mathbb{R}^n$ and a baseline input $x' \in \mathbb{R}^n$ (typically a zero vector). Integrated Gradients compute the attributions of the input relative to this baseline.

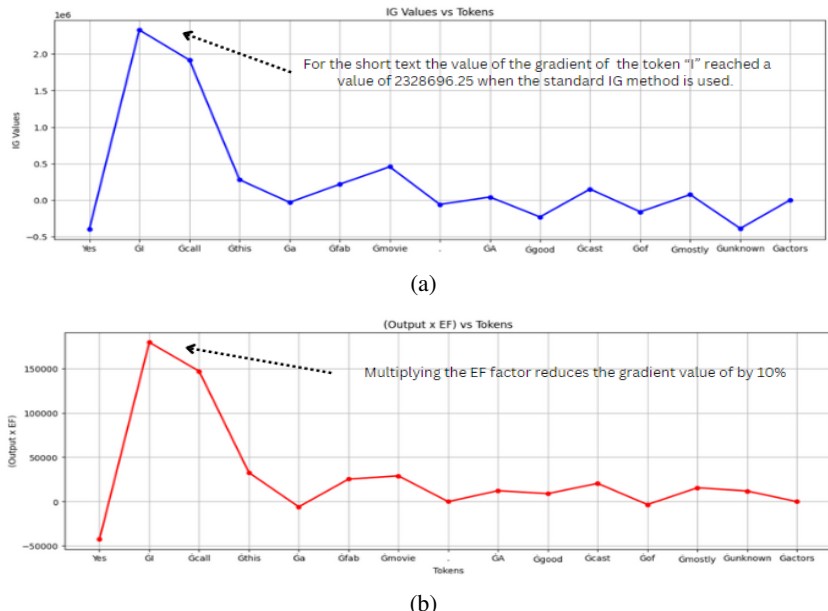

(a)

(b)

Figure 2: (a) shows the gradient values of the previously generated tokens with respect to the "actors" token calculated by the Integrated Gradient method. As the distance between the tokens increases, the gradient value increases and may explode if the text under consideration is very long. (b) shows the gradient values of the previously generated tokens with respect to the "actors" token calculated by our proposed method AIEG. The method has scaled down the gradient values by about 10%. The EF factor helps to capture only the gradients where the model makes decisions.

The attribution of the $i$-th word is given by:

$$\text{IG}_i(x) = (x_i - x_i') \int_{\alpha=0}^{1} \frac{\partial F(x' + \alpha(x - x'))}{\partial x_i} \, d\alpha \tag{3}$$

This formula represents the integrated gradient along the straight line between the baseline $x'$ and the input $x$.

Although the method has demonstrated effectiveness in neural machine translation, it fails to address certain limitations specific to generation tasks, which we will explore in the following sections.

### 3.1 LIMITATIONS IN INTEGRATED GRADIENT METHOD

While the method outlined in this paper has proven effective in numerous applications, it encounters specific challenges when applied to long text generation with auto-regressive models such as GPTs. Firstly, the method struggles with problems similar to exploding and vanishing gradient issues that arise when computing the gradients of the output relative to input tokens, as shown in figure 1(c). Secondly, it neglects the impact of attention—a critical component in large language models (LLMs)—in the attribution calculations of input tokens concerning the output tokens. Thirdly, this method assigns equal importance to all gradients, even in regions where the model's decision remains unchanged, leading to the accumulation of low-quality gradients. The issue of exploding and vanishing gradients is presented below as a theorem.

**Theorem:** *Consider an auto-regressive neural network represented by the function $F : \mathbb{R}^n \to \vec{e}$, where $\vec{e}$ is the embedding word vector of dimension $n$. Given an input sequence $\mathbf{x}_1, \mathbf{x}_2, \ldots, \mathbf{x}_T$ where $\mathbf{x}_t \in \mathbb{R}^n$ is the input vector at time step $t$, the hidden state of $F$ is updated iteratively at each time step. Denote the hidden state at time step $t$ as $\mathbf{h}_t \in \mathbb{R}^m$, where $m$ is the dimension of the hidden layer. We propose that when long sequences of text are considered $(T >> 1)$, the calculation of Integrated Gradients may result in undefined or numerically unstable values during the calculation*

of $\frac{\partial y_t}{\partial x_{t'}}$, where $y_t$ is the output generated at time $t$ and $x_{t'}$ is the input word that was generated at time $t'$.

**Proof:** Consider the scenario where we aim to calculate the attributions of the input words for the output word generated at time $t$. In autoregressive models—such the output at time $t-1$ serves as input for generating the output at time $t$. Given Equations (1) and (2), the gradient of the output at time step $t$, denoted as $y_t$, with respect to an input word $x_{t'}$ at time step $t'$, derived from an interpolated input, is expressed as follows. **For $t' = t$ (the same time step):**

The gradient of the output $y_t$ with respect to the input $x_t$ is:

$$\frac{\partial y_t}{\partial x_t} = \frac{\partial y_t}{\partial h_t} \cdot \frac{\partial h_t}{\partial x_t} \tag{4}$$

Where:

$$\frac{\partial y_t}{\partial h_t} = \mathbf{W}_y \cdot \sigma'(\mathbf{W}_y h_t + \mathbf{b}_y) \tag{5}$$

$$\frac{\partial h_t}{\partial x_t} = \mathbf{W}_x \cdot \sigma'(\mathbf{W}_h h_{t-1} + \mathbf{W}_x x_t + \mathbf{b}_h) \tag{6}$$

**For $t' < t$ (previous time steps):**

The gradient of the output $y_t$ with respect to an earlier input $x_{t'}$ (where $t' < t$) requires us to account for the effect of $x_{t'}$ on all subsequent hidden states up to $h_t$:

$$\frac{\partial y_t}{\partial x_{t'}} = \frac{\partial y_t}{\partial h_t} \cdot \frac{\partial h_t}{\partial h_{t-1}} \cdot \frac{\partial h_{t-1}}{\partial h_{t-2}} \cdots \frac{\partial h_{t'+1}}{\partial h_{t'}} \cdot \frac{\partial h_{t'}}{\partial x_{t'}} \tag{7}$$

Where:

$$\frac{\partial h_t}{\partial h_{t-1}} = \mathbf{W}_h \cdot \sigma'(\mathbf{W}_h h_{t-1} + \mathbf{W}_x x_t + \mathbf{b}_h) \tag{8}$$

and $\sigma'$ is the derivative of the activation function.

Based on equations 7 and 8, it is clear that the gradients of each hidden layer are successively multiplied by the gradients of the previous layers, leading to an accumulation of gradients during the computation. Following this, the Integrated Gradients (IG) values are determined using equation 3. This process, particularly for long sequences, can result in gradient explosion due to the cumulative summation, as depicted in figure 2, which may introduce instability in the gradient calculations. $\square$

## 4 OUR PROPOSED METHOD

### 4.1 MOTIVATION

We aim to identify the positive contributions of individual input tokens towards the generation of the output token. In this work, we address the limitations of Integrated Gradients (IG) discussed in the previous section. Our approach aims to mitigate the risk of gradient explosion by scaling down the value of $\frac{dF}{dx}$ and focusing only on high-quality gradients, i.e., those where the logits exhibit rapid changes (Walker et al. (2024)). By selectively considering these gradients, we effectively reduce the likelihood of gradient explosion in $\frac{dF}{dx}$. Also, given the critical role that attention mechanisms play in Large Language Models (LLMs) for next-token generation, it is essential to incorporate attention weights when calculating attributions. The attention mechanism, introduced by Vaswani (2017), allows models to focus on relevant parts of an input sequence, emphasizing key information when processing long sequences, shown in figure 1(b), and improving performance in tasks like translation, summarization, and question-answering. Consequently, attention is crucial for LLMs' decision-making in the next token generation and should be considered alongside integrated gradients in attribution calculations. Additionally, large language models like GPTs (Radford et al., 2019) use a technique known as masked self-attention, which plays a pivotal role in sequence generation tasks.

**Axiom: Attention:** *Consider two words, $\mathbf{x}_{t1}$ and $\mathbf{x}_{t2}$, within a sequence of words in a sentence, with attention values $A_{t1,t}$ and $A_{t2,t}$, respectively, corresponding to the generated output word at time $t$. If $A_{t1,t} > A_{t2,t}$, then it implies that the word $\mathbf{x}_{t1}$ has a greater impact and contribution towards the*

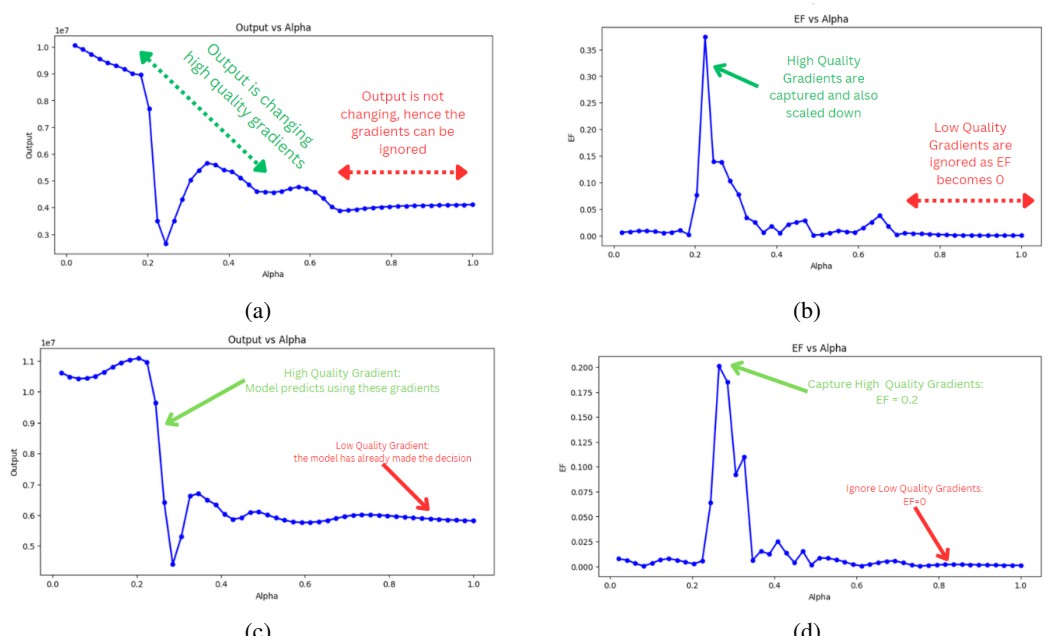

Figure 3: The above diagrams illustrate the Output Vs Alpha and EF Vs Alpha graphs of two different texts. (a) and (c) illustrates how the output changes with varying values of $\alpha$. Here the output is the L1 normalisation of the embedding vector of the targeted output. It is evident that around $\alpha = 0.8$, the model makes its prediction, and beyond this point, the model maintains its decision. (b) and (d) shows the variation of the emphasis factor (EF) with respect to the values of $\alpha$. Our method focuses on high-quality gradients, where the model makes decisions (rapid change of output), while in regions of low-quality gradients, the EF becomes 0, reducing the entire term in our proposed method to 0.

*generation of the output word at time $t$. This higher attention value reflects that $\mathbf{x}_{t1}$ is considered more relevant and influential in the context of the output prediction compared to $\mathbf{x}_{t2}$.* □

Previous attribution methods have largely overlooked the significance of attention mechanisms in their computations. In contrast, we propose using the aforementioned axiom to calculate the attribution of each input word towards the output. Let for an auto-regressive model with $L$ decoder layers and $H$ multi-headed masked self-attention mechanisms, Attention$_{t',t,h,l}$ denote the attention value of an input word $x_{t'}$ generated at time $t'$ with respect to the output word $x_t$ generated at time $t$ (where $t' < t$), for the $l^{th}$ layer and $h^{th}$ attention head. The overall attention of that input word towards the output word $NetAttention_{t',t}$ can then be computed as:

$$\text{NetAttention}_{t',t} = \frac{1}{L} \sum_{l=0}^{L} \left( \frac{1}{H} \sum_{h=0}^{H} \text{Attention}_{t',t,h,l} \right) \tag{9}$$

where,

$$\text{Attention}_{t',t,h,l} = \left( \text{softmax} \left( \frac{\mathbf{Q}_{t',h,l} \mathbf{K}_{t,h,l}^T}{\sqrt{d_k}} \right) \right) \mathbf{V}_{t',h,l} \tag{10}$$

$\mathbf{Q}_{t',h,l}$ is the query vector of token $x_{t'}$, $\mathbf{K}_{t,h,l}$ is the key vector of token $x_t$, $\mathbf{V}_{t',h,l}$ is the value vector of the $x_{t'}$ token for the $l^{th}$ layer and the $h^{th}$ attention head and $\mathbf{d}_k$ is the dimensionality of the key/query vectors.

Building upon the above theorem and axiom, we introduce our proposed attribution method, Attended Integrated and Emphasized Gradients (AIEG), along with the Emphasis Factor (EF). Consider an auto-regressive model generating a token $x_t$. Our goal is to compute the attribution of a previously generated token $x_{t'}$, where $0 < t' < t$. In this context, $x'_t$ serves as the baseline for the input $x_t$, and

$$\text{AIEG}_{t'}(x_t) = NetAttention_{t',t} \times \text{PosNorm}\left( (x_{t'} - x'_{t'}) \times \int_{\alpha=0}^{1} \frac{\partial F(x'_t + \alpha(x_t - x'_t))}{\partial x_{t'}} \times \mathbf{EF} \, d\alpha \right)$$
(11)

where,

$$\text{EF} = \frac{|F(x'_t + \alpha(x_t - x'_t)) - F(x'_t + (\alpha - \epsilon)(x_t - x'_t))|}{|F(x'_t + \alpha(x_t - x'_t))| + |F(x'_t + (\alpha - \epsilon)(x_t - x'_t))|},$$
(12)

$$\text{PosNorm}(a) = \frac{a}{\sum_{T=1}^{t-1} \text{AIEG}_T(x_t)},$$
(13)

where, $a > 0$, $\forall T \ \text{AIEG}_T(x_t) > 0$, $\epsilon$ is the minimum difference between two values of $\alpha$, $(\alpha - \epsilon) \geq 0$ and $0 < \epsilon < 1$. Also $||F(x'_t + \alpha(x_t - x'_t))| + |F(x'_t + (\alpha - \epsilon)(x_t - x'_t))|| > 0$, so that the EF remains defined for all values of $x_t, x'_t$ and $\alpha$.

The PosNorm function normalizes the attribution of a token generated at time $t'$ for the output token at time $t$ ($t' < t$) across all positive attributions of tokens generated from $T = 1$ to $T = t - 1$. We focus solely on positive attributions as we are interested in identifying words that positively contribute to the output. In our approach, attention and gradients are weighted equally, as both contribute equally to the generation of the next token. If the gradient value is high but the attention value of the input token with respect to the output token is low, the overall attribution decreases, and the reverse is also true. This has been depicted in figure 1 (a) with the text "The movie took me by surprise." and the output word with respect to which the gradients are calculated is "surprise". The algorithm for the method has been discussed in Appendix A.3. Next, we will present two theorems, along with their proofs, and three axioms to further explore the properties of the above equations.

**Theorem1:** *Consider a function $F(x) : \mathbb{R} \to \mathbb{R}$ and the Emphasis Factor EF function, mentioned in equation 12, which is continuous over the entire range of $F(x)$. We argue that $F(x) \times EF \leq F(x)$, $\forall F(x) \in \mathbb{R}$, keeping the sign of $F(x)$ intact.*

**Proof:** The Emphasis Factor (EF) can be expressed in a simple form as $\text{EF} = \frac{|m-n|}{|m|+|n|}$.

**When $m \neq n$:** $\forall m$ and $n$, where $m \neq n$, the following holds: $0 < \text{EF} \leq 1$. This is true because $|m - n| = |m + (-n)| \leq |m| + |-n| = |m| + |n|$ by the triangle inequality.
**When $m = n$:** we have: $\text{EF} = 0$

Thus, in both cases, the product $F(x) \times \text{EF}$ satisfies the following condition: $F(x) \times \text{EF} \leq F(x)$. This demonstrates that the Emphasis Factor ensures the product is always less than or equal to the original function $F(x)$ and hence checks the exploding gradient that arises while calculating the gradients. Also, since $EF$ is always greater than equal to zero, it keeps the sign of the product the same as $F(x)$. Hence the contribution of the words remains same, that is, a positively contributing word does not change to negative because of EF.$\square$

**Theorem2:** *Consider a function $F(x) : \mathbb{R} \to \mathbb{R}$ and the Emphasis Factor EF function, We assert that the Emphasis Factor prioritizes gradients in regions where the model is making decisions, while disregarding gradients in areas where the output has already been predicted.*

**Proof:** Consider an input token with an attention value greater than 0 with respect to the output token. When $F(x'_t + \alpha(x_t - x'_t)) \neq F(x'_t + (\alpha - \epsilon)(x_t - x'_t))$, the model is still in the decision-making phase, resulting in EF $> 0$, and thus $\frac{\partial F(x'_t + \alpha(x_t - x'_t))}{\partial x'_t} \times \text{EF} \, d\alpha > 0$. Conversely, when $F(x'_t + \alpha(x_t - x'_t)) = F(x'_t + (\alpha - \epsilon)(x_t - x'_t))$, the model has already made a decision, implying EF $= 0$, and $\frac{\partial F(x'_t + \alpha(x_t - x'_t))}{\partial x'_t} \times \text{EF} \, d\alpha = 0$ for a specific value of $\alpha$. Hence, the Emphasis Factor (EF) selectively captures only high-quality gradients, filtering out low-quality ones. Figure 3 and Appendix A.7 shows this theorem through graphs for different examples.$\square$

Due to the properties outlined in Theorems 1 and 2, the use of the EF effectively mitigates the gradient explosion issue commonly observed in standard Integrated Gradients.

| | GPT2-small | | | GPT-nano | | | LLaMA | | |
|--------|-------|--------|--------|-------|--------|--------|--------|--------|--------|
| **Method** | **LO↓** | **Comp↑** | **Suff↓** | **LO↓** | **Comp↑** | **Suff↓** | **LO↓** | **Comp↑** | **Suff↓** |
| Grad*Inp | -0.245 | 0.173 | 0.322 | -0.290 | 0.165 | 0.368 | -0.360 | 0.148 | 0.445 |
| IG | -0.527 | 0.338 | 0.260 | -0.780 | 0.362 | **0.236** | -1.180 | 0.310 | 0.415 |
| IGCG | -0.480 | 0.278 | 0.174 | -0.435 | 0.229 | 0.280 | -1.040 | 0.295 | 0.418 |
| DeepLIFT | -0.195 | 0.054 | 0.488 | -0.299 | 0.079 | 0.433 | -0.174 | 0.064 | 0.469 |
| GradShap | -0.377 | 0.217 | 0.309 | -0.522 | 0.167 | 0.346 | -0.685 | 0.224 | 0.434 |
| Attn.-Only | -0.137 | 0.121 | 0.294 | -0.144 | 0.133 | 0.308 | -0.177 | 0.187 | 0.445 |
| **AIEG** | **-0.535** | **0.348** | **0.140** | **-0.860** | **0.368** | 0.258 | **-1.510** | **0.395** | **0.355** |

Table 1: Comparison of our proposed method with various feature attribution methods across three language models fine-tuned and tested on the SST-2 dataset. For ↑ metrics, higher values indicate better performance, while for ↓ metrics, lower values are preferred.

| | GPT2-small | | | GPT-nano | | | LLaMA | | |
|--------|-------|--------|--------|-------|--------|--------|--------|--------|--------|
| **Method** | **LO↓** | **Comp↑** | **Suff↓** | **LO↓** | **Comp↑** | **Suff↓** | **LO↓** | **Comp↑** | **Suff↓** |
| Grad*Inp | -0.252 | 0.170 | 0.319 | -0.115 | 0.163 | 0.370 | -0.233 | 0.145 | 0.442 |
| IG | -0.530 | 0.334 | 0.165 | -0.792 | 0.358 | 0.254 | -1.185 | 0.305 | 0.413 |
| IGCG | -0.482 | 0.276 | 0.179 | -0.429 | 0.227 | 0.284 | -1.048 | 0.291 | 0.412 |
| DeepLIFT | -0.196 | 0.073 | 0.487 | -0.198 | 0.080 | 0.432 | -0.175 | 0.065 | 0.470 |
| GradShap | -0.478 | 0.216 | 0.310 | -0.521 | 0.168 | 0.347 | -0.684 | 0.225 | 0.365 |
| Attn.-Only | -0.138 | 0.111 | 0.295 | -0.143 | 0.134 | 0.309 | -0.176 | 0.186 | 0.446 |
| **AIEG** | **-0.542** | **0.345** | **0.137** | **-0.865** | **0.365** | **0.240** | **-1.515** | **0.393** | **0.351** |

Table 2: Comparison of our proposed method with various feature attribution methods across three language models fine-tuned and tested on the IMDB dataset. For ↑ metrics, higher values indicate better performance, while for ↓ metrics, lower values are preferred.

**Axiom: Sensitivity:** *Consider an autoregressive neural network function $F$, which is continuous and differentiable with respect to $\alpha$, ensuring that $\frac{\partial F}{\partial \alpha}$ is well-defined. Our attribution method at $\alpha$ along a given path is defined as: $\frac{\partial F}{\partial x} \times EF$. The term $(x - x')$ is omitted here, as it is a post-processing factor. For an input token closely related to the output token $NetAttention > 0$.*
*When $EF = 0$, which implies the change in output is $0$ and therefore, the attribution is naturally zero. Conversely, when $EF \neq 0$, at least one feature will have $\frac{\partial F}{\partial x} \neq 0$, resulting in a nonzero attribution. Therefore, by definition, our proposed attribution method AIEG satisfies the Sensitivity axiom.□*

**Axiom: Implementation Invariance:** *Consider an autoregressive neural network $F$, where $g$ is the input at time $t'$, $h$ represents the hidden layer, and $f$ is the output generated at time $t$ (with $t > t'$). In AIEG, the computation of $\frac{\partial f}{\partial g}$ is performed through the chain rule, such that $\frac{\partial f}{\partial g} = \frac{\partial f}{\partial h} \cdot \frac{\partial h}{\partial g}$. Given that the input word $g$ contributes positively to the output token ($NetAttention > 0$), our proposed method adheres to the principle of Implementation Invariance.* □

**Axiom: Linearity:** *Assume we combine two autoregressive deep networks, represented by the functions $f_1$ and $f_2$, to form a third network that models the function $a \times f_1 + b \times f_2$, i.e., a linear combination of the two networks. The attributions computed by the AIEG method for $a \times f_1 + b \times f_2$ result in a weighted sum of the attributions for $f_1$ and $f_2$, with weights $a$ and $b$, respectively. Therefore, our method satisfies the principle of linearity.* □

## 5 EXPERIMENT AND EVALUATION

### 5.1 EXPERIMENT DESIGN

We evaluate our proposed method against the following baseline models: Grad*Inp (Shrikumar et al. (2016)), Integrated Gradients (IG) (Sundararajan et al. (2017)), Integrated Gradients with Clipped Gradients (IGCG) as described in Appendix A.2, DeepLift (Shrikumar et al. (2017)), GradientShap

In a land of endless battles and ancient kingdoms, there lived warrior named Kaelaris with countless aliens for more than fifty millennia who waged an epic war between powers as many foes in every galaxy have joined together over the age.There exist only

(a) These are the attributions calculated by AIEG method where the word of interest is "galaxy".

In a land of endless battles and ancient kingdoms, there lived warrior named Kaelaris with countless aliens for more than fifty millennia who waged an epic war between powers as many foes in every galaxy have joined together over the age.There exist only

(b) These are the attributions calculated by IG method where the word of interest is "galaxy".

In the heart of a bustling city, where towering skyscrapers reached for sky and streets to turn white ancient was often one places with most spectacular views as commercial district. While others looked on with curiosity, this city also had capital, and the surrounding center home to the entire family. The most notable aspect of this village its rich diverse population. Over centuries it been site many legends and legends. The legends of ancient cities came out nowhere: about power status gods demons giants devils

(c) These are the attributions calculated by AIEG method where the word of interest is "cities".

In the heart of a bustling city, where towering skyscrapers reached for sky and streets to turn white ancient was often one places with most spectacular views as commercial district. While others looked on with curiosity this also had capital surrounding center home entire family. The most notable aspect of this village its rich diverse population. Over centuries it been site many legends and legends. The legends of ancient cities came out nowhere: about power status gods demons giants devils

(d) These are the attributions calculated by IG method where the word of interest is "cities".

One clear night as the world slept a bright light appeared over small town. I looked out window of house in which it had appeared. A huge sign had been added to top building. Inside the sign was picture an elf and her father whom I never met before. My dad spent summer after my parents separated and married in North America. He had gone to college at Yale. His name was William C "Bill" McHenry

(e) These are the attributions calculated by AIEG method where the word of interest is "married".

One clear night, as the world slept a bright light appeared over small town. I looked out window of house in which it had appeared. A huge sign had been added to top building. Inside the sign was picture an elf and her father whom I never met before. My dad spent summer after my parents separated and married North America. He had gone to college at Yale His name William C "Bill" McHenry

(f) These are the attributions calculated by IG method where the word of interest is "married".

On a quiet afternoon, Ella found an old, dusty key hidden beneath the floorboard urn and began to open it. She'd just been told that lock on door and lock in the window was broken

(g) These are the attributions calculated by AIEG method where the word of interest is "window".

On a quiet afternoon, Ella found an old, dusty key hidden beneath the floorboard urn and began to open it. She'd just been told that lock on door and lock in the window was broken

(h) These are the attributions calculated by IG method where the word of interest is "window".

Figure 4: Here, we compare the outputs generated by the AIEG and IG methods. In both cases, words are color-coded, with greener words indicating higher attribution toward the target word. It is evident that the AIEG method highlights words that carry more meaningful contributions, whereas the IG method produces less interpretable attributions.The IG attributions lack clarity. The above texts are generated by prompting GPT2-small and then their attributions are calculated using the two methods.

(Lundberg (2017)) and Attention-Only method (here we consider on the self-attention values of the input tokens with respect to the token of interest). For benchmarking, we employ the Stanford Sentiment Treebank (SST2)(Socher et al. (2013)) and IMDB (Maas et al. (2011)) datasets, comparing performance across the GPT2-small, GPT-nano, and Llama (Touvron et al. (2023)) models using the following metrics:

- **Log-odds (LO) score**: Shrikumar et al. (2017), measures the average change in negative logarithmic probabilities for the predicted class when the top k% of features are masked using zero padding. Lower scores indicate better performance.
- **Comprehensiveness (Comp) score:** DeYoung et al. (2020), quantifies the average change in predicted class probability resulting from the removal of the top k% of features. A higher score indicates better performance.
- **The Sufficiency (Suff) score:** DeYoung et al. (2020), measures the average change in predicted class probability when only the top k% of features are retained. This score evaluates how well the top k% attributions alone account for the model's prediction.

In our study, we consider $\hat{y}$ as the predicted output token at time step $t$ for a given input. To assess model performance, we will remove the top $k\%$ (in our case, the value of k is 20%) of the words

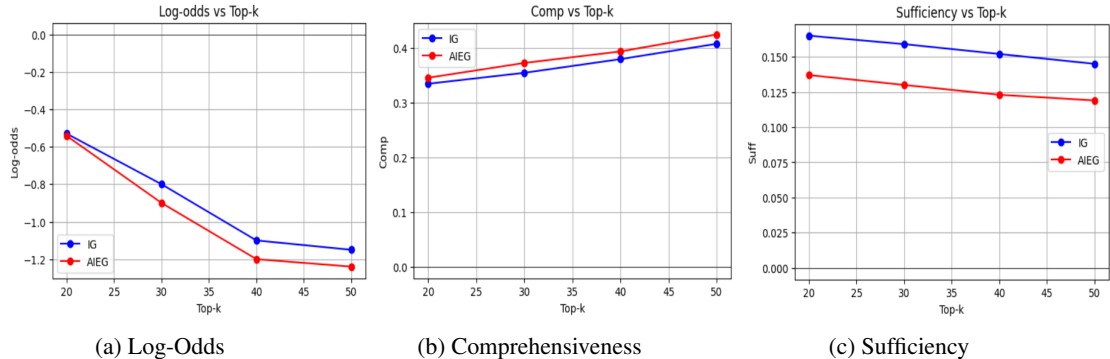

(a) Log-Odds            (b) Comprehensiveness            (c) Sufficiency

Figure 5: Impact of varying the top-k% on the log-odds, comprehensiveness, and sufficiency metrics for the GPT2-small model fine-tuned on the IMDb (Maas et al. (2011)) dataset.

predicted by the respective models. This approach will provide insight into the models' confidence and facilitate a comparative performance analysis. For the Integrated Gradients with Clipped Gradients (IGCG) method, we applied a threshold of $1,000,000$ to clip extreme gradient values during the attribution calculation. Detailed explanations of the metrics has been discussed in Appendix A.4.

SST2 contains 11,855 individual sentences extracted from movie reviews, while the IMDB dataset consists of 50,000 movie reviews. We randomly selected 5000 reviews from each dataset and fine-tuned the models as masked language models. A smaller number of examples was chosen for fine-tuning, as our objective is to understand the model's behaviour rather than to generate high-quality, task-related outputs. Similarly for testing, we randomly selected around 2100 movie reviews from each dataset and used a portion of the review to construct a paragraph of 50, 200, and 400 tokens, with each category having an equal amount of movie reviews (700). Since the model outputs tokens, we convert them back to words before presenting the final output. For words that are split during tokenization, the tokens are reassembled, and their individual attributions are summed to compute the attribution of the entire word. From the generated text, we manually selected a token of interest to calculate its positive attribution based on the preceding tokens. The attributions were computed and compared across different models. Table 1 presents the attribution comparisons for the SST2 dataset, while Table 2 compares the results for the IMDB dataset.

## 5.2 RESULTS

Tables 1 and 2 compare the performance of our proposed algorithm against the other attribution methods discussed above. Our results consistently outperform the other methods across the datasets and language models. This suggests that the attention mechanism and the emphasis factor play a crucial role in determining the attribution of each token towards the output token. In Figures 4, we aim to identify the positive attribution of the input words toward the output word (the word of interest) and compare the outputs generated by our proposed method (left examples) with those from Integrated Gradients (right examples). The greener a word appears, the greater its positive contribution to the word of interest. In Figures 4 (a) and (b), the word of interest is "galaxy." It is clear that the words with the highest attributions in (a) are "land," "lived," "warrior," "aliens," and "wars," which are coherent. In contrast, (b), generated by the IG method, highlights "In" along with the other words. Similarly, from the other examples, it is evident that our proposed method outperforms IG. More visual examples has been shown in Appendix A.6, where we have compared the attributions computed by all the above-mentioned methods. We tested our method with text summarising and compared it with the IG method in Appendix A.5. In almost all the cases AIEG gives more reasonable attributions than IG method.

**Ablation Studies on the values of k in Evaluation Metrics**: Figure 5 illustrates the impact of varying the top-k% on the log-odds, comprehensiveness, and sufficiency metrics for the GPT2-small model fine-tuned on the IMDb dataset. We compare our AIEG method against the Integrated Gradients (IG). Our results show that both variants outperform IG across all values of k. Notably, the performance gap between AIEG and IG is minimal at lower k values but progressively widens as k

increases, as depicted in figure 5 (a). In figure 5 (b) and figure 5 (c) the gap between the values remains almost the same but AIEG outperforms IG for all values of $k$.

## 6 CONCLUSION

In this paper, we demonstrated the limitations of the Integrated Gradients (IG) method in computing input token attributions toward the output token. Specifically, we highlighted the issue of exploding gradients when calculating the gradients of input tokens with respect to the output. To address this, we introduced the Attended Integrated and Emphasized Gradients (AIEG) method, which mitigates the exploding gradient problem by focusing on high-quality gradients. Our proposed method consistently outperforms other approaches in attribution calculation across multiple datasets and models.

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

# A  APPENDIX

## A.1  RELATED WORKS

Explainability in machine learning, particularly for large language models (LLMs), has become a crucial area of research as these models grow in complexity. The "black-box" nature of models like GPT and LLaMA-2 Touvron et al. (2023) poses significant challenges in understanding how these models make predictions, leading to a demand for more transparent methods to interpret their behaviour.

Local explainability techniques such as SHAP Lundberg (2017) and LIME Ribeiro et al. (2016) have been widely adopted to provide insight into the contributions of individual input features. These methods rely on perturbations and attribution techniques to assess the influence of tokens on model outputs. However, they are often computationally intensive and assume feature independence, which may not hold in real-world datasets Feng et al. (2018). Also, these methods may not be able to capture the decision of why the model generated the token under consideration. Gradient-based methods such as Integrated Gradients Sundararajan et al. (2017) accumulate the gradients along the input feature path, providing a smoother attribution but at the cost of higher computational demand and reduced faithfulness ( Sikdar et al. (2021); Shrikumar et al. (2017)).

Global explainability focuses on extracting and interpreting broader patterns within models. Probing-based methods have been essential in identifying the syntactic and semantic representations encoded within LLMs (Hewitt & Manning (2019); Peng et al. (2022)). Studies by Geva et al. (2022) and Kobayashi et al. (2023) delve into the internal mechanisms of models, showing that feed-forward networks and attention heads capture complex linguistic knowledge. Mechanistic interpretability has also become an essential field, aiming to reverse-engineer neural networks into comprehensible circuits Wang et al. (2022), allowing for a deeper understanding of tasks like indirect object identification.

Model editing has recently garnered attention as a way to directly alter specific knowledge within LLMs without extensive retraining. Techniques such as hypernetwork-based editing Mitchell et al. (2022) and causal tracing Meng et al. (2022)) allow for targeted interventions in model behavior, improving its responses to particular inputs. These techniques have shown potential in enabling models to adapt without disrupting overall performance Yao et al. (2023).

Explainability has also been used to enhance task-specific capabilities. In-context learning (ICL), for instance, has benefited from studies showing that specific attention heads play a pivotal role in transferring knowledge from prompt examples to downstream tasks (Hendel et al. (2023); Todd et al. (2023)). Moreover, explainability methods like inference-time intervention (ITI) have been leveraged to address issues of hallucination in text generation, where models generate outputs that deviate from factual content. Li et al. (2024) demonstrated that truthful interventions in attention layers could significantly enhance the factuality of model outputs, mitigating the impact of hallucinations.

Beyond improving factuality, explainability has also been used to tackle biases within models. Techniques like integrated gradients (Sundararajan et al. (2017)) and its variations have been applied to identify neurons responsible for social biases (Liu et al. (2024)), offering a pathway to fairer and more ethically aligned language models.

Overall, the body of research highlights the importance of developing both local and global explainability methods to improve trust and transparency in LLMs. These methods not only facilitate understanding but also open new avenues for enhancing the performance and ethical alignment of models in diverse NLP applications.

## A.2  WHY NOT USE GRADIENT CLIPPING BEYOND A THRESHOLD TO STOP THE GRADIENT EXPLOSION?

We are suggesting that instead of using an emphasis factor, when the gradient exceeds a predefined positive threshold, further multiplication is halted. This approach prevents the gradients from becoming too small or too large, ensuring more stable and meaningful gradient calculations. Specifically, we define positive and negative thresholds $\theta^+$ and $\theta^-$, respectively. The gradients are modified as follows:

$$\frac{\partial h_t}{\partial h_{t-1}}^{\text{(capped)}} = \begin{cases} \frac{\partial h_t}{\partial h_{t-1}}, & \text{if } \frac{\partial h_t}{\partial h_{t-1}} \leq \theta^+ \\ \theta^+, & \text{if } \frac{\partial h_t}{\partial h_{t-1}} > \theta^+ \end{cases}$$

Clipping the gradients has a significant impact on the calculation of attributions. This approach may hinder the proper accumulation of critical gradients, particularly in areas where the model makes key decisions. As a result, the method fails to accurately highlight the tokens that contribute the most to the output. The impact of gradient clipping on the attribution process is further illustrated in the accompanying figures in Appendix A.6.

## A.3 PROPOSED ALGORITHM

1. Encode input text to token IDs: $x = \text{tokenizer(text)}$
2. Set baseline: $x_0 = 0$ (if not provided)
3. Initialize total gradients: $G_{\text{total}} = 0$
4. For each $\alpha \in [0, 1]$ with steps $N$ :

    $x_i(\alpha) = x_0 + \alpha(x - x_0)$

    $y(\alpha) = \text{model}(x_i(\alpha))$

    $\nabla x_i(\alpha) = \dfrac{\partial y(\alpha)}{\partial x_i(\alpha)}$

    $\Delta G(\alpha) = \nabla x_i(\alpha) \cdot \dfrac{\left| y(\alpha) - y(\alpha - \frac{1}{N}) \right|}{|y(\alpha)| + |y(\alpha - \frac{1}{N})|}$

    $G_{\text{total}} + = \Delta G(\alpha)$

    $G_{\text{avg}} = \dfrac{G_{\text{total}}}{N}$

5. Compute IG scores: $IG(x) = (x - x_0) \cdot G_{\text{avg}}$
6. Normalize IG scores: $IG_{\text{norm}}(x) = \dfrac{IG(x)}{\sum IG(x)}$
7. Extract attention and Average: $A = $ average attention from each layer $l$ and head $h$
8. Compute contribution: $C(x) = IG_{\text{norm}}(x) \cdot A$
9. Return token contributions: $C(x)$ for each token

## A.4 EVALUATION METRICS

- **Log-odds (LO) score**: Shrikumar et al. (2017), measures the average change in negative logarithmic probabilities for the predicted class when the top k% of features are masked using zero padding. To calculate this, the top k% of words are identified based on attribution scores from an explanation algorithm and are then replaced with zero padding. Specifically, for a dataset with $N$ sentences, the LO score is defined as:

$$\text{log-odds}(k) = \frac{1}{N} \sum_{i=1}^{N} \log \left( \frac{p(\hat{y} \mid x_i^{(k)})}{p(\hat{y} \mid x_i)} \right)$$

where $\hat{y}$ is the predicted class, $x_i$ is the $i$-th sentence, and $x_i^{(k)}$ is the modified sentence with the top k% words replaced by zero padding. Lower scores indicate better performance.

- **Comprehensiveness (Comp) score:** DeYoung et al. (2020), quantifies the average change in predicted class probability resulting from the removal of the top k% of features. This score, similar to the Log-odds, assesses the impact of the most influential words on the model's prediction. It is defined as:

$$\text{Comp}(k) = \frac{1}{N} \sum_{i=1}^{N} \left[ p(\hat{y} \mid x_i^{(k)}) - p(\hat{y} \mid x_i) \right]$$

where $x_i^{(k)}$ represents the modified sentence with the top k% of words removed. Higher scores indicate better performance.

- **The Sufficiency (Suff) score:** DeYoung et al. (2020), measures the average change in predicted class probability when only the top k% of features are retained. This score evaluates how well the top k% attributions alone account for the model's prediction. It is calculated similarly to the Comprehensiveness score, but here $x_i^{(k)}$ refers to the sentence containing only the top k% of words. Lower scores indicate better performance.

## A.5 APPLICATIONS IN OTHER TASKS

We evaluated our model for text summarization. For this purpose, we employed the XSum dataset (Narayan et al. (2018)). The GPT-2 (small) model was used for summarization, with the string "TL;DR" appended to the end of the input to guide the summarization process. Following the generation of the summarized text, we computed attributions for each word and aggregated their contributions. For example, given the summarized text "The scientists discovered a new animal," we determined the contribution of each word, starting from "The" to "animal," and accumulated their respective contributions. Since it is an autoregressive model, the words generated in the summarised text depend on the previous words as well, and therefore, even the summarised text has its contributions in generating the next word. This process is illustrated in Figures 6 to 14 below.

We then compared the output of the Integrated Gradients (IG) method with that of the AIEG method. Our results indicate that the attributions produced by AIEG are more reasonable and consistent compared to those from IG.

A species of crustacean which lives in the gut of sea cucumbers has been discovered in the waters off Scotland. Scientists found the "furry" crab, which is about 3cm long, during a research project in the Inner Hebrides. The creature has been named after its thick coating of hair, which gives it a striking appearance. It is also known to feed on the mucus and faeces of sea cucumbers. The researchers described it as a "strange and fascinating" species.

Summary:
A new species of furry crab has been found in Scottish waters.

**Output from AIEG**

A species of crustacean which lives in the gut of sea cucumbers has been discovered in the waters off Scotland . Scientists found the "furry" crab , which is about 3cm long, during a research project in the Inner Hebrides. The creature has been named after its thick coating of hair, which gives it a striking appearance. It is also known to feed on the mucus and faeces of sea cucumbers. The researchers described it as a "strange and fascinating" species

Summary:
A new species of furry crab has been found in Scottish waters.

**Output from IG**

A species of crustacean which lives in the gut of sea cucumbers has been discovered in the waters off Scotland. Scientists found the "furry" crab, which is about 3cm long, during a research project in the Inner Hebrides. The creature has been named after its thick coating of hair, which gives it a striking appearance. It also known to feed on the mucus and faeces of sea cucumbers . The researchers described it as a "strange and fascinating" species

Summary:
A new species of furry crab has been found in Scottish waters.

Figure 6: Image 1

A new study has found that extreme weather conditions could lead to a spike in premature births. Researchers analysed data from more than 2.3 million births across 50 US states between 2000 and 2010. They found that an increase in the number of days with extremely hot temperatures was linked to a rise in premature deliveries. The research, published in the journal Environment International, calls for urgent action to address the impact of climate change on health. A study suggests extreme weather may cause more premature births in the US.

Summary
A study suggests extreme weather may cause more premature births in the US.

**Output from IG**

A new study has found that extreme weather conditions could lead to a spike in premature births . Researchers analysed data from more than 23 million births across 50 US states between 2000 and
2010. They found that an increase in the number of days with extremely hot temperatures was linked to a rise in premature deliveries. The research , published in the journal Environment International, calls for urgent action to address the impact of climate change on health.

Summary
A study suggests extreme weather may cause more premature births in the US.

**Output from AIEG**

A new study has found that extreme weather conditions could lead to a spike in premature births . Researchers analysed data from more than 23 million births across 50 US states between 2000 and
2010. They found that an increase in the number of days with extremely hot temperatures was linked to a rise in premature deliveries. The research , published in the journal Environment International, calls for urgent action to address the impact of climate change on health.

Summary:
A study suggests extreme weather may cause more premature births in the US.

Figure 7: Image 2

Parents who send their children to private schools may be wasting their money, according to a new report. Researchers claim that a state school education is just as likely to produce high-achieving pupils as private schooling. The study examined the academic achievements of more than 4,000 pupils at both types of school. It found that any advantage from private schooling "disappears" by the time pupils reach the age of 16. A report suggests state schools are just as good as private schools at producing high achievers.

**Output from AIEG**

Parents who send their children to private schools may be wasting their money , according a new report . Researchers claim that a state school education is just as likely to produce high-achieving pupils as private schooling. The study examined the academic achievements of more than 4000 pupils at both types of school. It found that any advantage from private schooling "disappears " by the time reach the age of 16.

A report suggests state schools are just as good as private schools at producing high achievers.

**Output from IG**

Parents who send their children to private schools may be wasting their money , according a new report . Researchers claim that a state school education is just as likely to produce high -achieving pupils as private schooling. The study examined the academic achievements of more than 4000 pupils at both types of school. It found that any advantage from private schooling "disappears " by the time reach the age of 16.

A report suggests state school are just as good as private schools at producing achievers

Figure 8: Image 3

A man who set up a fake hospital to offer illegal abortions has been sentenced to three years in prison. The man, known as Dr. Henry, had operated the clinic in Lagos for several years. Police said he lured women into the clinic by promising low-cost medical procedures. The court heard how he charged up to $1,000 for each abortion and often left women with life-threatening complications.

Summary:
A man in Lagos has been jailed for running a fake clinic offering illegal abortions.

**Output from AIEG**

A man who set up a fake hospital to offer illegal abortions has been sentenced to three years in prison . The man, known as Dr Henry, had operated the clinic in Lagos for several years. Police said he lured women into the clinic by promising low -cost medical procedures. The court heard how he charged up to $1000 for each abortion and often left women with life-threatening complications.

Summary:
A man in Lagos has been jailed for running a fake clinic offering illegal abortions

**Output from IG**

A man who set up a fake hospital to offer illegal abortions has been sentenced to three years in prison . The man, known as Dr Henry, had operated the clinic in Lagos for several years. Police said he lured women into the clinic by promising low -cost medical procedures. The court heard how he charged up to $1000 each abortion and often left women with life threatening complications.

Summary:
A man in Lagos has been jailed for running a fake clinic offering illegal abortions

Figure 9: Image 4

The first images of the inside of a nuclear fusion experiment have been released by scientists. The pictures show the inside of the "tokamak" reactor at the UK Atomic Energy Authority's facility in Oxfordshire. The device is being used in research aimed at harnessing the power of nuclear fusion to generate electricity. It is hoped that this form of energy will one day provide a clean, virtually limitless source of power.

Summary:
Scientists have released the first images of a nuclear fusion experiment in the UK.

**Output from AIEG**

The first images of the inside of a nuclear fusion experiment have been released by scientists . The pictures show the inside of the "tokamak " reactor at the UK Atomic Energy Authority 's facility in Oxfordshire. The device is being used in research aimed at harnessing the power of nuclear fusion to generate electricity. It is hoped that this form of energy will one day provide a clean , virtually limitless source of power.

Summary:
Scientists have released the first images of a nuclear fusion experiment in the UK.

**Output from IG**

The first images of the inside of a nuclear fusion experiment have been released by scientists . The pictures show the inside of the "tokamak " reactor at the UK Atomic Energy Authority's facility in Oxfordshire. The device is being used in research aimed at harnessing the power of nuclear fusion to generate electricity. It is hoped that this form of energy will one day provide a clean , virtually limitless source of power.

Summary:
Scientists have released the first images of a nuclear fusion experiment in the UK.

Figure 10: Image 5

The number of homeless people in the UK has risen by 10% in the last year, according to government statistics. Charities have warned that the situation could worsen if measures aren't taken to provide more affordable housing.

Summary:
Homelessness in the UK has increased by 10% over the past year.

**Output from AIEG**

The number of homeless people in the UK has risen by 10 % in the last year , according to government statistics . Charities have warned that the situation could worsen if measures aren 't taken to provide more affordable housing.

Summary:
Homelessness in the UK has increased by 10% over the past year

**Output from IG**

The number of homeless people in the UK has risen by 10 % in the last year , according to government statistics . Charities have warned that situation could worsen if measures aren 't taken to provide more affordable housing.

Summary:
Homelessness in the UK has increased over the past year

Figure 11: Image 6

A new study has shown that regular physical activity can help reduce the risk of heart disease. The research, conducted over a 10-year period, found that people who exercised for at least 30 minutes a day had a significantly lower risk of developing heart-related conditions.

Summary:
Regular exercise can lower the risk of heart disease, according to a study.

**Output from AIEG**

A new study has shown that regular physical activity can help reduce the risk of heart disease . The research , conducted over a 10 -year period, found that people who exercised for at least 30 minutes a day had a significantly lower risk of developing heart-related conditions.

Summary:
Regular exercise can lower the risk of heart disease, according to a study

**Output from IG**

new study has shown that regular physical activity can help reduce the risk of heart disease . The research , conducted over a 10 -year period, found that people who exercised for at least 30 minutes a day had significantly lower risk of developing heart-related conditions.

Summary:
Regular exercise can lower the risk of heart disease, according to a study

Figure 12: Image 7

The local council has announced plans to build a new community park in the center of town. The park will feature playgrounds, walking paths, and areas for outdoor sports. Construction is expected to begin later this year.

Summary:
A new community park is set to be built in the town center, according to the local council.

**Output from AIEG**

The local council has announced plans to build a new community park in the center of town . The park will feature playgrounds , walking paths and areas for outdoor sports . Construction is expected to begin later this year.

Summary:
A new community park is set to be built in the town center, according to the local council

**Output from IG**

The local council has announced plans to build a new community park in the center of town . The park will feature playgrounds , walking paths and areas for outdoor sports. Construction is expected to begin later this year.

Summary:
A new community park is set be built in the town center, according to the local council.

Figure 13: Image 8

A fire broke out at a historic building in downtown London late last night. Firefighters worked through the night to contain the blaze, and no injuries have been reported. Investigators are now trying to determine the cause of the fire.

Summary:
A fire at a historic building in London has been brought under control.

**Output from AIEG**

A fire broke out at a historic building in downtown London late last night . Firefighters worked through the night to contain blaze , and no injuries have been reported. Investigators are now trying to determine the cause of the fire.

Summary:
A fire at a historic building in London has been brought under control

**Output from IG**

A fire broke out at a historic building in downtown London late last night . Firefighters worked through the night to contain blaze , and no injuries have been reported Investigators are now trying to determine the cause of the fire.

Summary:
A fire at a historic building in London has been brought under control

Figure 14: Image 9

## A.6 Visual Comparisons among the Methods

In this section, we present additional visual examples comparing the attribution calculations for each method. We employed the GPT-2 (small) model, finetuned with the IMDb dataset Prerak et al. (2023) using 500 data points, and generated text based on small prompts. The generated sequences varied in length, randomly set to 50, 100, or 200 tokens. We then calculated the attributions for specific words of interest, as illustrated in Figures 15 to 29. The colour intensity of green reflects the magnitude of each word's contribution—darker green indicates a higher contribution. Across almost all examples, the AIEG method produced more interpretable and reasonable attributions compared to the IG method.

Figure 15: Image 1

Input Prompt:
A married daughter who longs for her mother's love is visited by the latter, a successful concert pianist.

Generated Text:
A married daughter who longs for her mother 's love is visited by the latter , a successful concert pianist . The couple have no children and all but four are raised his wife But their lives diverge sharply when they realize their mother.
Word of Interest --> mother

**Output from AIEG**

A married daughter who longs for her mother 's love is visited by the latter , a successful concert pianist . The couple have no children and all but four are raised his wife But their lives diverge sharply when they realize their mother

**Output from IG**

A married daughter who longs for her mother 's love is visited by the latter , a successful concert pianist . The couple have no children and all but four are raised his wife But their lives diverge sharply when they realize their mother

**Output from Inp*Grad**

A married daughter who longs for her mother 's love is visited by the latter , a successful concert pianist . The couple have no children and all but four are raised his wife But their lives diverge sharply when they realize their mother

**Output from IGCG**

A married daughter who longs for her mother 's love is visited by the latter , a successful concert pianist . The couple have no children and all but four are raised his wife But their lives diverge sharply when they realize their mother

**Output from DeepLift**

A married daughter who longs for her mother 's love is visited by the latter , a successful concert pianist . The couple have no children and all but four are raised his wife But their lives diverge sharply when they realize their mother

**Output from GradShap**

A married daughter who longs for her mother ' love is visited by the latter , a successful concert pianist . The couple have no children and all but four are raised his wife But their lives diverge sharply when they realize their mother

**Output from Attention-Only**

A married daughter who longs for her mother ' love is visited by the latter , a successful concert pianist . The couple have no children and all but four are raised his wife But their lives diverge sharply when they realize their mother

Figure 16: Image 2

Input Prompt:
A simple act of kindness always sparks another, even in a frozen, faraway place.

Generated Text:
A simple act of kindness always sparks another , even in a frozen faraway place . This was probably the case for most humans who encountered me too The fact that I would willingly let them go through my pain and anger.
Word of Interest --> anger

**Output from AIEG**

A simple act of kindness always sparks another , even in a frozen faraway place | This was probably the case for most humans who encountered me too The fact that I would willingly let them go through my pain and anger

**Output from IG**

A simple act of kindness always sparks another , even in a frozen faraway place . This was probably the case for most humans who encountered me too The fact that I would willingly let them go through my pain and anger

**Output from Inp*Grad**

A simple act of kindness always sparks another , even in a frozen faraway place . This was probably the case for most humans who encountered me too The fact that I would willingly let them go through my pain and anger

**Output from IGCG**

A simple act of kindness always sparks another , even in a frozen faraway place . This was probably the case for most humans who encountered me too The fact that I would willingly let them go through my pain and anger

**Output from DeepLift**

A simple act of kindness always sparks another , even in a frozen faraway place . This was probably the case for most humans who encountered me too The fact that I would willingly let them go through my pain and anger

**Output from GradShap**

A simple act of kindness always sparks another , even in a frozen faraway place . This was probably the case for most humans who encountered me too The fact that I would willingly let them go through my pain and anger

**Output from Attention-Only**

simple act of kindness always sparks another , even in a frozen faraway place . This was probably the case for most humans who encountered me too The fact that I would willingly let them go through my pain and anger

Figure 17: Image 3

Input Prompt:
A pragmatic U.S. Marine observes the dehumanizing effects the Vietnam War

Generated Text:
A pragmatic U .S Marine observes the dehumanizing effects Vietnam War and other wars on American veterans He calls it "War On Drugs ." It has been a war heart ," says veteran who served in was.
Word of Interest --> veteran

**Output from AIEG**

A pragmatic U .S Marine observes the dehumanizing effects Vietnam War and other wars on American veterans He calls it "War On Drugs ." It has been a war heart , says veteran who served in was

A pragmatic U.S. Marine observes the dehumanizing effects the Vietnam War and other wars on American veterans. He calls it the "War On Drugs." It has been a "war on the heart," says the veteran who served in Vietnam and was

**Output from IG**

A pragmatic U .S Marine observes the dehumanizing effects Vietnam War and other wars on American veterans He calls it "War On Drugs ." says veteran who served in was

**Output from Inp*Grad**

A pragmatic U .S Marine observes the dehumanizing effects Vietnam War and other wars on American veterans He calls it "War On Drugs ." It has been a war heart ," says veteran who served in was

**Output from IGCG**

A pragmatic U .S Marine observes the dehumanizing effects Vietnam War and other wars on American veterans He calls it "War On Drugs ." It has been a war heart ," says veteran who served in was

**Output from DeepLift**

A pragmatic U .S Marine observes the dehumanizing effects Vietnam War and other wars on American veterans He calls it "War On Drugs ." It has been a war heart ," says veteran who served in was

**Output from GradShap**

A pragmatic U .S Marine observes the dehumanizing effects Vietnam War and other wars on American veterans He calls it "War On Drugs ." It has been a war heart ," says veteran who served in was

**Output from Attention-Only**

pragmatic U .S Marine observes the dehumanizing effects Vietnam War and other wars on American veterans He calls it "War On Drugs ." It has been a war heart ," says veteran who served in was .

Figure 18: Image 4

Figure 19: Image 5

Figure 20: Image 6

Figure 21: Image 7

Figure 22: Image 8

Figure 23: Image 9

Figure 24: Image 10

Figure 25: Image 11

Prompt: When the menace known as the Joker wreaks havoc and chaos on the people of Gotham, Batman must accept one of the greatest psychological and physical tests of his ability to fight injustice
Generated Text:
When the menace known as the Joker wreaks havoc and chaos on the people of Gotham, Batman must accept one of the greatest psychological and physical tests of his ability to fight injustice. In a small New York neighbourhood to protest the recent government decree prohibiting crime, Bane and the city band meet on the street and

Word of Interest: crime

**Output from AIEG**

When the menace known as Joker wreaks havoc and chaos on the people of Gotham, Batman must accept one greatest psychological and physical tests of his ability to fight injustice. In a small New York neighbourhood to protest the recent government decree prohibiting crime, Bane and the city band meet on the street and

**Output from IG**

When the menace known as Joker wreaks havoc and chaos on the people of Gotham, Batman must accept one greatest psychological and physical tests of his ability to fight injustice. In a small New York neighbourhood to protest the recent government decree prohibiting crime, Bane and the city band meet on the street and

**Output from Inp*Grad**

When the menace known as Joker wreaks havoc and chaos on people of Gotham , Batman must accept one greatest psychological physical tests his ability to fight injustice . In a small New York neighborhood protest recent government decree prohibiting crime Bane city band meet street

**Output from IGCG**

When the menace known as Joker wreaks havoc and chaos on people of Gotham , Batman must accept one greatest psychological physical tests his ability to fight injustice . In a small New York neighborhood protest recent government decree prohibiting crime Bane city band meet street

**Output from DeepLift**

When the menace known as Joker wreaks havoc and chaos on people of Gotham , Batman must accept one greatest psychological physical tests his ability to fight injustice . In a small New York neighborhood protest recent government decree prohibiting crime Bane city band meet street

**Output from GradShap**

When the menace known as Joker wreaks havoc and chaos on people of Gotham , Batman must accept one greatest psychological physical tests his ability to fight injustice . In a small New York neighborhood protest recent government decree prohibiting crime Bane city band meet street

**Output from Attention-Only**

When the menace known as Joker wreaks havoc and chaos on people of Gotham , Batman must accept one greatest psychological physical tests his ability to fight injustice . In a small New York neighbourhood protest recent government decree prohibiting crime Bane city band meet street

Figure 26: Image 12

Figure 27: Image 13

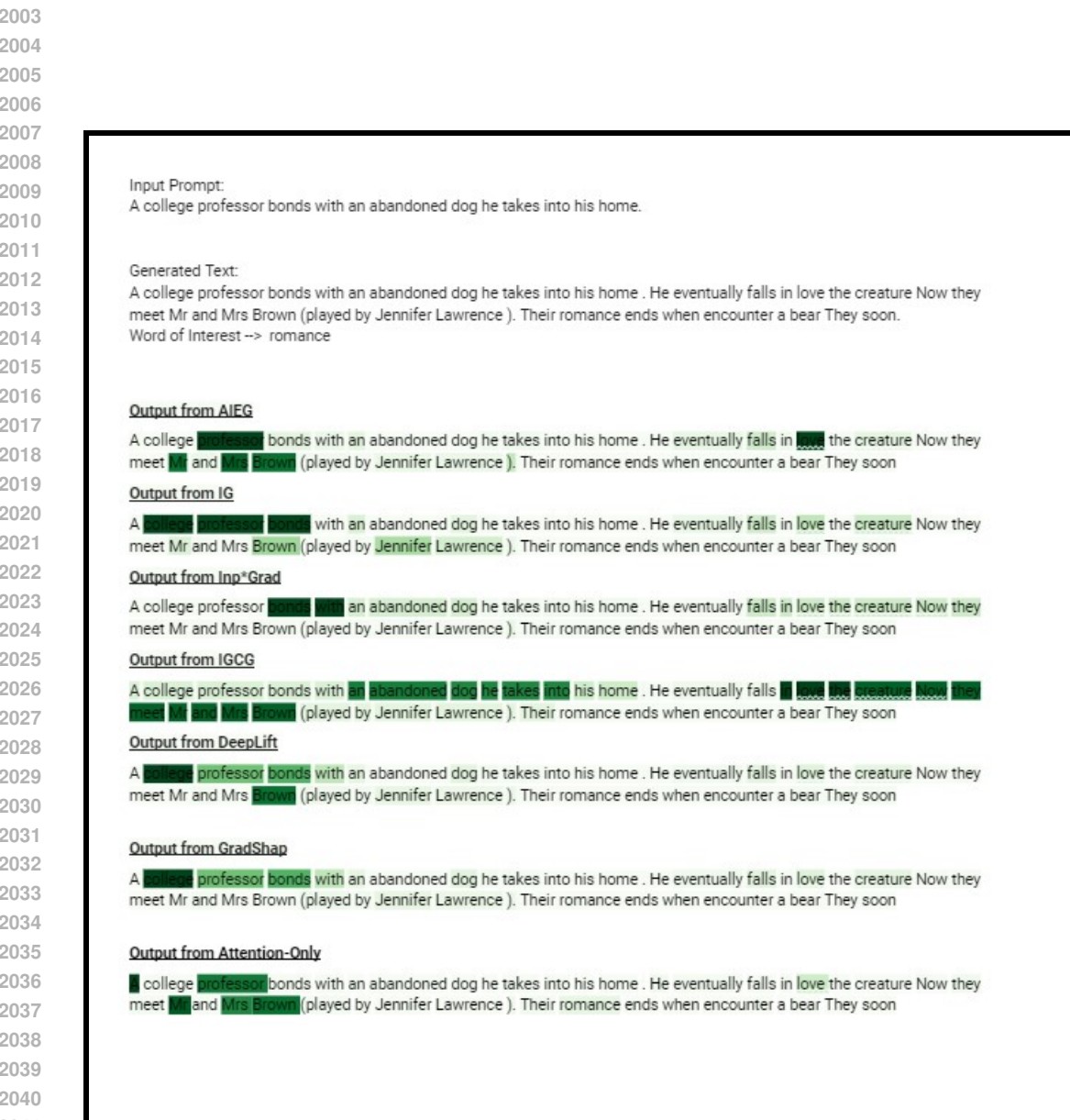

Figure 28: Image 14

Figure 29: Image 15

## A.7 GRAPHICAL VISUALISATION OF THE EF AND AIEG VALUES

In this section, Figures 30 to 34 present graphical representations of various metrics: IG values vs. Tokens, EF (Exponential Factor) vs. Alpha, Output vs. Alpha, (Output x EF) vs. Tokens, and AIEG values vs. Tokens for short sentences. These visualizations provide valuable insights into the behavior of our proposed method and highlight the key differences compared to the standard IG approach.

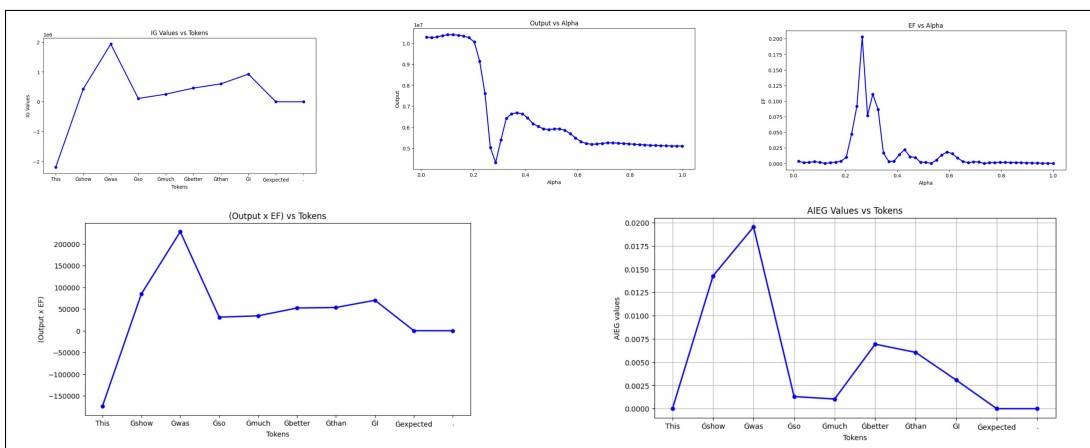

Figure 30: Images 1

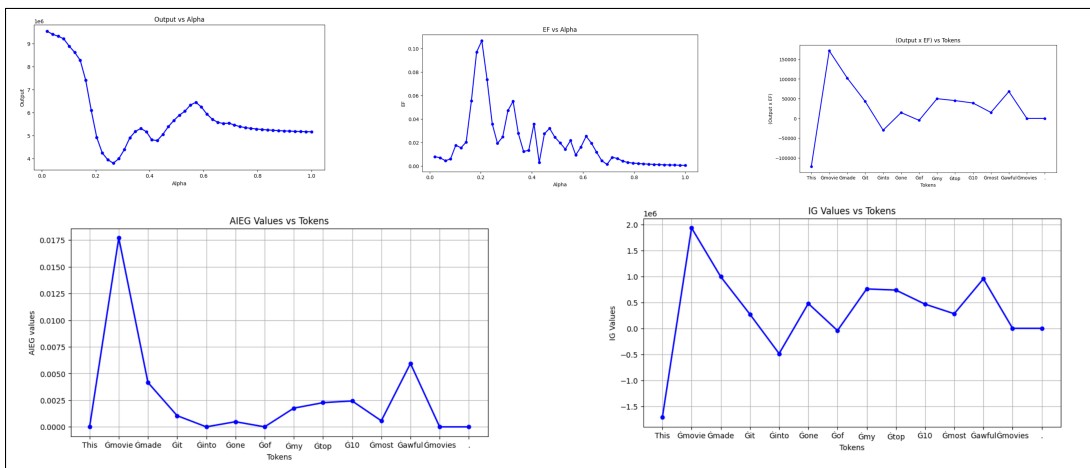

Figure 31: Images 2

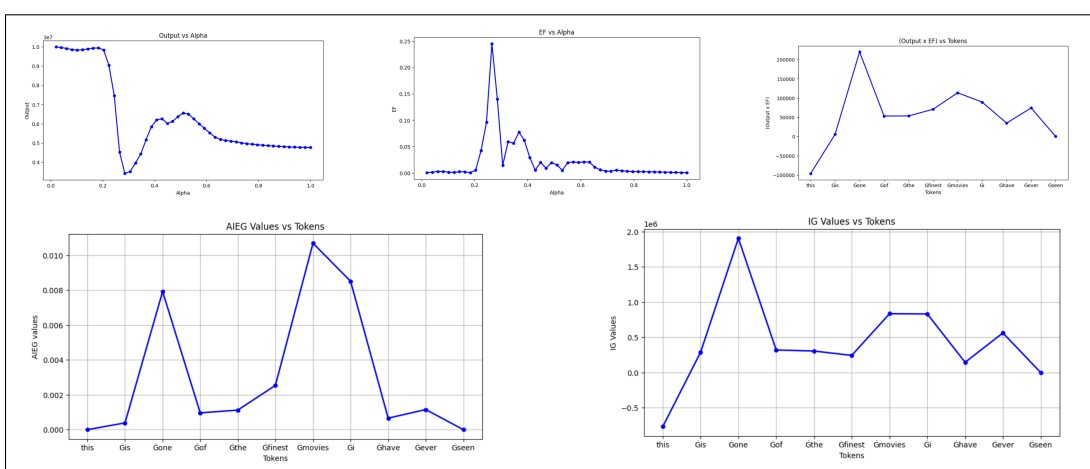

Figure 32: Images 3

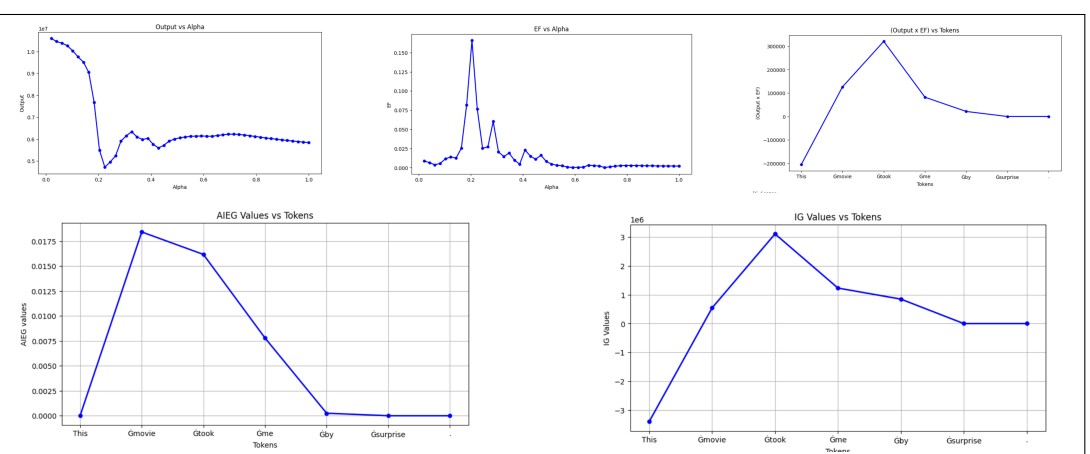

Figure 33: Images 4

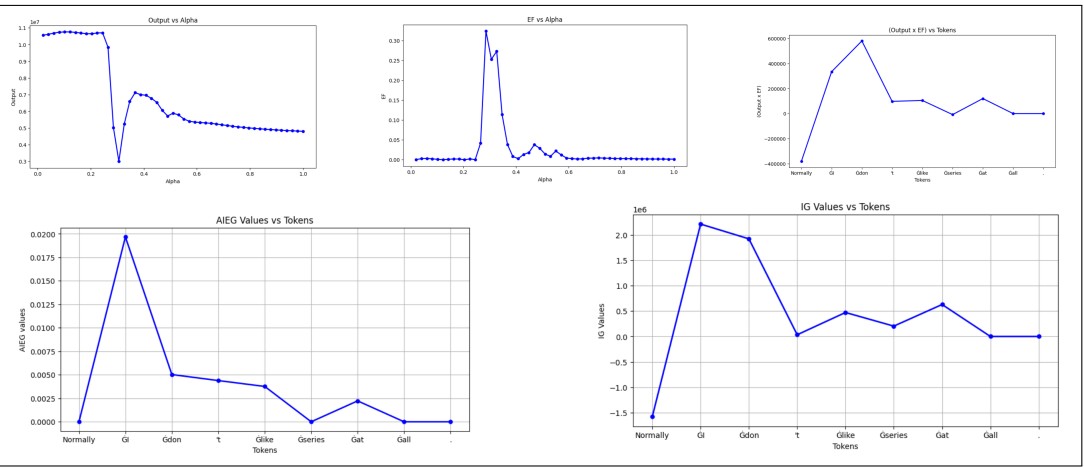

Figure 34: Images 5

