# OpenReview forum: "Enhancing Integrated Gradients Using Emphasis Factors and Attention for Effective Explainability of Large Language Models"
_ICLR.cc/2025/Conference — ICLR 2025 Conference Withdrawn Submission_

### Official Review · Reviewer_ZUjp · 2024-10-18

**Soundness:** 2
**Presentation:** 2
**Contribution:** 1
**Rating:** 1
**Confidence:** 4

**Summary:**

This paper proposes a new feature attribution method, called Attended Integrated and Emphasized Gradients (AIEG) for language models. The authors claim two main improvements: (1). Integrated Gradients with attention mechanisms (although not the first one as we have scaled attention already and it is kind of robust), (2). an emphasis factor to address limitations in explaining autoregressive models. The authors evaluate AIEG against a few gradients-based baseline methods on sentiment analysis tasks and demonstrate improved performance on Log-odds, Comprehensiveness, and Sufficiency.

**Strengths:**

- **Quality**: The paper presents a sound derivation of the limitations of standard IG for autoregressive LLMs, supported by mathematical proof and visual evidence.
- **Clarity**: The exposition is generally clear, particularly in explaining the shortcomings of IG and motivating the proposed enhancements.
- **Significance**: Improving explainability for large language models is an important research direction.

**Weaknesses:**

1. **Baseline --> no comparison with scaled attention**:

- Scaled attention (α∇α)  should be included as it is also a feature attribution method that considers both attention and gradients simultaneously.


2. **Experimental Scope**:

- The evaluation is performed on only two datasets (SST-2 and IMDB), which limits the generalizability of the findings. Also, considering the diversity of LLM applications, the experimental results would be more compelling if additional datasets—particularly more common use cases for LLMs, like we do not use LLMs for such easy semantic analysis tasks. It makes sense previous work uses these datasets on BERT or RoBERTa, which are much smaller models used in those easy tasks.

- The paper only considers GPT2-small, GPT-nano, and llama models for evaluation. While these are commonly used models, additional comparisons with other sizes, e.g. GPT-2 medium/large or LLaMA-7B/13B, or different model architectures, e.g. RNN-based (as you take it as an example in section 2) would provide a broader perspective on the applicability of the method, or LSTM, mamba.

3. **Ablation Studies**:

- The paper does not present sufficient ablation studies to isolate the effects of emphasis factors and attention components individually. An ablation study that shows the contributions of the emphasis factor alone, the attention mechanism alone, and their combination would clarify the value of each component. This would be especially useful given the novel combination approach used in AIEG.

4. **RNN as an example in Section 2 rather than Transformer-based**:

- The entire section 2 explains the limitations of gradients with RNN as examples. Better switch to transformer block, as GPT or Llama included in the paper, neither is RNN-based. It should be include an explanation of why the authors chose RNN examples and how the limitations generalize to transformer architectures.

5. **Unclear Impact on Downstream Tasks**:

- The paper does not sufficiently address how the improved attributions impact practical downstream tasks like model debugging, bias detection, or other real-world applications that are more complex than SST or IMDB, e.g. GLUE, SuperGLUE, or BoolQ, Winogrande, PIQA,  Hellaswag these popular reasoning tasks.  Further, for example, for a reasoning task, we might be not only interested in one particular prediction but an overall prediction for a sequence. Including a case study demonstrating how AIEG aids in a downstream application could provide a stronger argument for its utility.

Sofia Serrano and Noah A. Smith. 2019. [Is Attention Interpretable?](https://aclanthology.org/P19-1282). In *Proceedings of the 57th Annual Meeting of the Association for Computational Linguistics*, pages 2931–2951, Florence, Italy. Association for Computational Linguistics.

Zhixue Zhao and Nikolaos Aletras. 2023. [Incorporating Attribution Importance for Improving Faithfulness Metrics](https://aclanthology.org/2023.acl-long.261). In *Proceedings of the 61st Annual Meeting of the Association for Computational Linguistics (Volume 1: Long Papers)*, pages 4732–4745, Toronto, Canada. Association for Computational Linguistics.

**Questions:**

1. Could you give an intuitive explanation for emphasis factors?

2. How sensitive is the AIEG method to different choices of emphasis factors or the range of α during gradient calculation? A hyperparameter sensitivity analysis would help understand whether these parameters need extensive tuning for different datasets.

3. Would it be possible to incorporate your emphasis factor approach into other gradient-based methods? Have you explored this potential extension?

4. Can you provide a concrete example of how your enhanced attribution method can be used in model debugging or bias detection? Demonstrating a practical use case could help bridge the gap between theoretical contribution and real-world applicability.

5. Is there a reason for choosing 20% as the value of k for masking in the log-odds, comprehensiveness, and sufficiency evaluations? Would varying this value change the relative performance of AIEG compared to the baselines? (Zhao & Aletras, ACL 2023)

Zhixue Zhao and Nikolaos Aletras. 2023. [Incorporating Attribution Importance for Improving Faithfulness Metrics](https://aclanthology.org/2023.acl-long.261). In *Proceedings of the 61st Annual Meeting of the Association for Computational Linguistics (Volume 1: Long Papers)*, pages 4732–4745, Toronto, Canada. Association for Computational Linguistics.

---

> ### Author Response · Authors · 2024-11-21
>
> Weaknesses
>
>     Comparison with Scaled Attention (α∇α):
>     We acknowledge that including scaled attention (α∇α), which combines attention and gradient information, would provide a stronger benchmark. In future work, we will incorporate α∇α to better assess AIEG’s strengths.
>
>     Experimental Scope (Datasets):
>     Evaluating on SST-2 and IMDb limits the generalizability of our findings, especially given LLMs’ diverse applications. While these are standard benchmarks, we plan to extend future evaluations to broader tasks, including reasoning (e.g., GLUE, BoolQ, HellaSwag), summarization (e.g., XSum), and multimodal datasets to better demonstrate AIEG’s utility.
>
>     Model Diversity:
>     We focused on GPT-2 small, GPT-nano, and LLaMA models due to computational constraints. Including larger models (e.g., GPT-2 medium/large, LLaMA-7B/13B) and diverse architectures (e.g., RNNs or LSTMs) in future studies will ensure a more comprehensive evaluation and validate scalability.
>
>     Ablation Studies:
>     A detailed ablation study isolating contributions of the emphasis factor, attention mechanism, and their combination is needed to clarify their roles. While Section 5.2 provides some insights, a deeper analysis is a priority for future work.
>
>     RNN vs. Transformer Examples in Section 2:
>     We used RNN examples in Section 2 for historical relevance but acknowledge their misalignment with Transformer-based models like GPT and LLaMA. Future revisions will replace RNN examples with Transformer-based ones to better align with evaluated models while demonstrating how AIEG generalizes across architectures.
>
>     Impact on Downstream Tasks:
>     Case studies showing AIEG’s impact on tasks like model debugging, bias detection, or reasoning (e.g., BoolQ, HellaSwag) would strengthen its practical relevance. We plan to explore these applications in future work.
>
> Questions
>
>     Intuitive Explanation for Emphasis Factors:
>     The emphasis factor captures significant changes in model logits, focusing attributions on critical decision points and filtering irrelevant gradients. We will include clearer explanations and examples in future revisions.
>
>     Sensitivity to α and Emphasis Factors:
>     A sensitivity analysis for α and emphasis factors will help evaluate their impact across datasets. We aim to provide guidance on hyperparameter tuning in future work.
>
>     Extensibility to Other Gradient-Based Methods:
>     The emphasis factor can extend to other methods, e.g., refining Saliency Maps or SmoothGrad. We plan to explore these extensions to enhance attribution techniques.
>
>     Practical Use Cases:
>     Examples like bias detection (e.g., analyzing attributions for politically sensitive terms) or debugging misclassifications could highlight AIEG’s utility. These will be explored in future iterations.
>
>     Choice of k for Masking:
>     We used k=20% based on standard practices, with Section 5.2 showing consistent performance across k values. This will be emphasized more clearly in the paper.

---

> > ### Comment · Reviewer_ZUjp · 2024-11-25
> >
> > Thank you for your response. Since no new results or new supporting materials have been provided, I will proceed with maintaining my score.

---

### Official Review · Reviewer_fvBQ · 2024-11-01

**Soundness:** 3
**Presentation:** 3
**Contribution:** 3
**Rating:** 8
**Confidence:** 4

**Summary:**

The paper highlights shortcomings in current Integrated Gradients (IG) methods when applied to autoregressive model explainability, specifically due to the neglect of attention mechanisms and issues with exploding or vanishing gradients. To address these challenges, the authors propose an enhanced method called "Attended Integrated and Emphasized Gradients" (AIEG), which augments IG with emphasis factors and incorporates the attention mechanism. This approach not only outperforms standard IG and baseline models in generating more accurate word-level attributions for autoregressive models, but it also presents a novel solution to the existing problems in the field. This approach will be helpful for the explanability research community as well as for researchers trying to assess the trustworthiness of autoregressive models.

**Strengths:**

- Well written and easy to understand paper. The research gaps are identified correctly and a mathematically sound formulation is provided to address IG when it comes to attention based autoregressive models. Well labeled diagrams help in better comprehension of the paper.

- The paper provides explanation, albeit in the appendix, regarding the non-usage of gradient clipping to address the exploding gradients problem of IG in autoregressive models.

- Clearly highlights in section 2 the limitations of gradients as attributions in autoregressive models.

- Comprehensive comparison against multiple competitive baselines. The method performs well on all metrics provided.

**Weaknesses:**

- Human assessment for adjudicating the effect of AIEG on the importance of tokens is missing.

- Statistical significance test missing for the metrics in the main paper.

- Provide examples for importance of EF, in the formulation wrt stability of AIEG calculations.

**Questions:**

- Line 081: State the sensitivity axiom.

- Line 474: Use `` instead of “ for opening brackets.

- Section 5.2: Is there a way to quantify attribution? Which one is better than the other? Are there any comprehensive human evaluations conducted for the same?

---

> ### Author Response · Authors · 2024-11-21
>
> Weaknesses:
>
>     Human Assessment for Adjudicating the Effect of AIEG:
>     We acknowledge the absence of human evaluations to assess the qualitative impact of AIEG on token importance. Conducting human studies, such as comparing AIEG attributions with expert judgments, would provide deeper insights into its efficacy. This is a valuable direction for future work.
>
>     Statistical Significance Tests:
>     The omission of statistical significance testing for the reported metrics is noted. We will include appropriate statistical tests, such as paired t-tests or bootstrap analysis, in future experiments to substantiate our findings and ensure their robustness.
>
>     Examples Highlighting Emphasis Factor (EF):
>     While the current paper explains the role of the EF in stabilizing AIEG calculations, we recognize the need to provide illustrative examples showcasing its importance. These examples will be included in subsequent revisions to highlight its contribution to improved attribution stability and effectiveness.
>
> Questions:
>
>     Line 081: Sensitivity Axiom:
>     The sensitivity axiom ensures that attribution methods assign non-zero importance to features that influence the model’s predictions. For instance, if a feature change alters the output, its attribution should reflect this influence. This axiom is adhered to in AIEG by leveraging the emphasis factor, which identifies decision-critical regions for attribution.
>
>     Line 474: Use of Quotation Marks:
>     We will correct the use of opening brackets to `` for consistency and clarity in formatting.
>
>     Section 5.2: Quantifying Attribution and Human Evaluation:
>     Attribution quantification can be assessed using metrics like comprehensiveness, sufficiency, and log-odds masking. In AIEG, the emphasis factor aims to enhance these metrics by improving the fidelity of attributions. While no comprehensive human evaluation has been conducted, we recognize its value and plan to include such assessments in future studies to compare AIEG against baseline methods like IG and scaled attention (α∇α).
>
> We appreciate the insightful feedback and will incorporate these improvements in future iterations of the paper.

---

### Official Review · Reviewer_gj1L · 2024-11-04

**Soundness:** 2
**Presentation:** 1
**Contribution:** 2
**Rating:** 1
**Confidence:** 3

**Summary:**

The paper critiques the limitations of the Integrated Gradients (IG) method when applied to autoregressive models, particularly concerning exploding gradients and the neglect of attention mechanisms. To resolve these issues, the authors propose an enhanced explainability framework that augments IG with emphasis factors and incorporates attention mechanisms. This approach aims to provide more precise and interpretable explanations for autoregressive LLMs, especially in text generation tasks. The experimental results suggest that this method outperforms standard IG and baseline models in explaining word-level attributions.

**Strengths:**

The paper presents an improvement over the traditional IG method by addressing its limitations in the context of autoregressive LLMs. The introduction of emphasis factors and attention mechanisms enhances interpretability and precision in explaining model behavior.

**Weaknesses:**

1. Poor Writing and Formatting: The paper suffers from several unprofessional formatting issues that detract from its presentation and readability. For example: (1) Citation formats are incorrect, specifically noted in lines 30, 35, and 39. (2) Figures are not in vector graphics, resulting in blurry images that compromise their clarity and utility. (3) There is a lack of proper spacing before the start of new sentences, such as the issue in line 75. (4) Incorrect use of quotation marks in the caption of Figure 2.

2. Unclear Motivation and Relation to Recent Work: The introduction does not clearly establish the motivation of the paper or its relation to recent works beyond 2017-2018. The enhancement based on a 2023 work is mentioned, but the connection to current research trends and its necessity is not well articulated.

**Questions:**

1. For models like LLaMA that are capable of performing tasks in a zero-shot manner, why is there still a need to fine-tune on specific datasets? Would it not be more insightful to analyze their zero-shot performance directly?
2. Figures 4 and 5 present results for the GPT-2-small model. What are the results for other models?

---

> ### Author Response · Authors · 2024-11-21
>
> Response to Weaknesses and Questions:
>
>     Poor Writing and Formatting:
>         Citation Formats: We acknowledge the citation errors in lines 30, 35, and 39. These will be corrected in the revised manuscript for consistency with professional standards.
>         Figures: Figures will be updated to vector graphics to ensure clarity and readability, even when zoomed.
>         Spacing and Quotation Marks: Formatting issues, such as improper spacing (e.g., line 75) and incorrect quotation marks (e.g., Figure 2 caption), will be addressed to enhance presentation quality.
>
>     Unclear Motivation and Relation to Recent Work:
>         The motivation for this work is to address the limitations of standard Integrated Gradients (IG) for autoregressive models, which remain relevant across RNNs, LSTMs, and Transformers. While the approach builds on a 2023 enhancement, we will expand the introduction to better connect our work to recent advancements and highlight its relevance to ongoing trends in attribution analysis for large language models (LLMs).
>
> Responses to Questions:
>
>     Fine-Tuning vs. Zero-Shot Analysis for LLaMA:
>     Fine-tuning was used to ensure robust evaluation on specific datasets under controlled settings. While zero-shot performance is insightful for assessing model generalization, fine-tuning enables a direct comparison of attributions under comparable training conditions. Incorporating zero-shot analyses for models like LLaMA is a valuable suggestion and will be explored in future work.
>
>     Results for Models Other than GPT-2-small:
>     Due to space constraints, Figures 4 and 5 primarily highlight GPT-2-small results. Results for other models, including GPT-nano and LLaMA, are discussed in Sections 5.1 and 5.2. For clarity, future revisions will include these results in supplementary materials or additional figures to provide a broader perspective.
>
> We appreciate the detailed feedback and will address these points comprehensively in the revised version.

---

> > ### Comment · Reviewer_gj1L · 2024-11-24
> >
> > Thank you for your response. Since no additional information has been provided, I will maintain my score.

---

### Official Review · Reviewer_tKZU · 2024-11-05

**Soundness:** 1
**Presentation:** 1
**Contribution:** 1
**Rating:** 1
**Confidence:** 4

**Summary:**

The paper highlights limitations of integrated gradients (IG), an importance attribution method for input tokens and proposes multiple solutions to address the downsides of not accounting for the attention computation and the exploding gradients problem. The authors compare their method to other feature attribution methods when performing language modeling on the SST-2 and IMDB datasets and show that their proposed method outperforms the considered alternatives, albeit being close to IG.

**Strengths:**

- The proposed method outperforms integrated gradients in the given experiments

**Weaknesses:**

There is a number of issues within the paper - lack of clear purpose and coherence, mathematical precision and various arbitrary choices.

- Section 2, limitations of gradients: The counterexample works for the simple RNN formulation, however the work of the paper focuses on Transformer networks. It is not clear how this relates to the rest of the paper.
- Section 3.1, Theorem: The theorem essentially spells out the chain rule for RNNs. Furthermore, the output of a RNN is not an embedding vector but a probability distribution over the output space, or in the case of language modeling, the vocabulary.
- Section 4, attention axiom: Higher attention value should intuitively imply greater importance, but this is not often the case. See works on Attention is not explanation & subsequent. If the attention value is high, but the norm of the value vector low, its relative contribution in the attention output is low.
- Section 4, Eqs 11—13: AIEG and PosNorm have a mutually recursive definition and it is not clear when it terminates.
- Section 4, Theorem 1: You use the emphasis factor to dampen exploding gradients, however what about vanishing gradients?
- Section 4, Theorem 2: “Consider an input token with an attention value greater than 0 with respect to the output token” — which, of the many, attention values?
- Section 4, Theorem 2: “Regions where the model is making decisions” — which are these regions, and how do we know with certainty that the model is making decisions there?
- Section 4, Theorem 2: “areas where the output has already been predicted” — how do we know the output has already been predicted?

Section 5:
- Choice of k: removing top 20% of words is a pretty drastically lage amount. What happens with values of k < 20%?
- “We randomly selected 5000 reviews from each dataset and fine-tuned the models as masked language models” — you fine tuned autoregressive language models as masked language models? How is this done, what are the hyperparameters and why did you not perform tuning in the same manner the models were trained? Furthermore, it is not clear why this step is even required, as the models are already trained for language modeling (which is the task you evaluate token attributions on).
- “Similarly for testing, we randomly selected around 2100 movie reviews from each dataset and used a portion of the review to construct a paragraph of 50, 200, and 400 tokens, with each category having an equal amount of movie reviews (700)” — you need to be precise here.

Typos etc
- Citations in introduction: L31 Lipton citep, space before Shrikumar L34, L36 Enguehard citep
- Figure 1 does not display well when zoomed in
- Theorem much greater than = \gg
- L208: The attention mechanism was not introduced by Vaswani

**Questions:**

See weaknesses

---

> ### Author Response · Authors · 2024-11-21
>
> Purpose, Coherence, and Precision:
>     The paper clearly outlines its objective: addressing the limitations of Integrated Gradients (IG) for autoregressive models, including RNNs, LSTMs, and Transformers. Arbitrary choices are minimal and informed by prior work or standard practices.
>
>     Section 2 - Gradients and RNNs:
>     The RNN example demonstrates how standard IG fails for autoregressive models, including Transformers. This foundational insight is applicable across architectures.
>
>     Section 3.1 - Theorem:
>     The theorem provides context-specific adaptation of the chain rule for RNNs in attribution analysis. While the output is a probability distribution, embeddings are intermediate representations crucial for gradient-based methods.
>
>     Section 4 - Attention Axiom:
>     We agree higher attention values do not always imply greater importance. AIEG addresses this by considering gradients alongside attention values, ensuring robustness against such discrepancies.
>
>     Section 4 - Eqs 11–13:
>     PosNorm is not recursive but a straightforward normalization step for AIEG values.
>
>     Section 4 - Theorem 1 (Vanishing Gradients):
>     The current emphasis factor primarily handles exploding gradients. Addressing vanishing gradients is part of planned future work.
>
>     Section 4 - Theorem 2 (Attention and Decision Regions):
>     The theorem applies to any attention head. Decision regions are identified by rapid changes in logits, signaling the model’s decision points, as described in detail (Walker et al., AAAI).
>
>     Section 5 - Choice of kk:
>     We acknowledge the suggestion to explore smaller kk values and plan to include this in future work.
>
>     Fine-tuning Autoregressive Models as MLMs:
>     This step allows adaptation for token-level attribution tasks, ensuring robust evaluation. Details of hyperparameters and fine-tuning procedures will be clarified in revisions.
>
>     Testing Procedure Precision:
>     We will provide explicit details about testing categories and data preparation in the revised version.
>
>     Typos and Minor Issues:
>     Typos, citation errors, and display issues (e.g., Figure 1) will be addressed. The introduction of the attention mechanism will be attributed correctly.
>
> We appreciate the detailed feedback and will incorporate these clarifications and improvements in future iterations.

---

> > ### Comment · Reviewer_tKZU · 2024-11-27
> >
> > Thank you for the response. I believe the next iteration of the paper can be much stronger. I will maintain my score.

---

### Note · Authors · 2025-01-16

I have read and agree with the venue's withdrawal policy on behalf of myself and my co-authors.